



# Interference from alkenes in chemiluminescent NOx measurements

Mohammed S. Alam[1], Leigh R. Crilley[1#], James D. Lee[2], Louisa J. Kramer[1],
Christian Pfrang[1], Mónica Vázquez-Moreno[3*], Amalia Muñoz[3], Milagros
Ródenas[3] and William J. Bloss[1]

[1] School of Geography, Earth and Environmental Sciences, University of Birmingham, Birmingham, UK
[2] National Centre for Atmospheric Science, Wolfson Atmospheric Chemistry Laboratories, University of York, York, UK;
[3] EUPHORE, Fundación CEAM, Valencia, Spain

\# now at: Department of Chemistry, York University, Toronto, ON, Canada
\* now at: FISABIO, Valencia, Spain

Correspondence to: m.s.alam@bham.ac.uk

## ABSTRACT

Nitrogen oxides ($NO_x$ = NO + $NO_2$) are critical intermediates in atmospheric chemistry. $NO_x$ levels control the cycling and hence abundance of the primary atmospheric oxidants OH and $NO_3$, and regulate the ozone production which results from the degradation of volatile organic compounds (VOCs) in the presence of sunlight. They are also atmospheric pollutants, and $NO_2$ is commonly included in air quality objectives and regulations. $NO_x$ levels also affect the production of the nitrate component of secondary aerosol particles and other pollutants such as the lachrymator peroxyacetyl nitrate (PAN). The accurate measurement of NO and $NO_2$ is therefore crucial to air quality monitoring and understanding atmospheric composition. The most commonly used approach for measurement of NO is chemiluminescent detection of electronically excited $NO_2$ ($NO_2^*$) from the NO + $O_3$ reaction. Alkenes, ubiquitous in the atmosphere from biogenic and anthropogenic sources, also react with ozone to produce chemiluminescence and thus may contribute to the measured $NO_x$ signal. Their ozonolysis reaction may also be sufficiently rapid that their abundance in the instrument background cycle, which also utilises reaction with ozone, differs from the measurement cycle – such that the background subtraction is incomplete, and an interference effect results. This interference has been noted previously, and indeed the effect has been used to measure both alkenes and ozone in the atmosphere. Here we report the results of a systematic investigation of the response of a selection of commercial $NO_x$ monitors, ranging from systems used for routine air quality monitoring to atmospheric research instrumentation, to a series of alkenes. Alkenes investigated range from short chain alkenes, such as ethene, to the biogenic monoterpenes. Experiments were performed in the European Photoreactor (EUPHORE) to ensure common calibration and samples for the monitors, and to unequivocally confirm the alkene levels present (via FTIR). The instrument interference responses ranged from negligible levels up to 11 % depending upon the alkene present and conditions used (*e.g.* presence of co-reactants and differing humidity). Such interferences may be of substantial importance for the interpretation of ambient $NO_x$ data, particularly for high-VOC, low-$NO_x$ environments such as forests, or indoor environments where alkene abundance from personal care and cleaning products may be significant.





## INTRODUCTION

Measurement of atmospheric trace constituents is central to atmospheric chemistry research and air pollution monitoring. Key challenges to trace measurements are sensitivity, reactivity and selectivity – as many components of interest are only present at ppb (parts per billion, $10^{-9}$) or ppt (parts per trillion, $10^{-12}$) mixing ratios; in many cases their inherent reactivity necessitates *in situ* detection, and as atmospheric trace composition comprises many thousands of different chemical components (Goldstein and Galbally, 2007). Consequently, specific measurement approaches have been developed to measure key atmospheric species, within the specific conditions (analyte abundance, presence of other constituents) anticipated (Heard, 2008). This paper reports a systematic study of the interference arising in measurements of the nitrogen oxides from the presence of alkenes in sampled air, when using the most widespread air quality monitoring technique of chemiluminescence detection.

$NO_x$ ($= NO + NO_2$) abundance controls the cycling and hence abundance of the primary atmospheric oxidants, hydroxyl (OH) and nitrate ($NO_3$) radicals, and regulates the ozone production which results from the degradation of volatile organic compounds (VOCs) in sunlight. $NO_x$ are also atmospheric pollutants in their own right, and $NO_2$ is commonly included in air quality objectives and regulations (as the more harmful component of $NO_x$) (European Environment Agency, 2018; Chaloulakou et al. (2008). In addition to their role in controlling ozone formation, $NO_x$ levels affect the production of other pollutants such as the lachrymator peroxyacetyl nitrate (PAN), and the nitrate component of secondary aerosol particles. Consequently, accurate measurement of nitrogen oxides in the atmosphere is of major importance for monitoring pollution levels and assessing consequent health impacts, and understanding atmospheric chemical processing. Atmospheric NO and $NO_2$ are formed from natural processes (lightning, soil emissions of NO, biomass burning and even snowpack emissions) and anthropogenic activities (high temperature combustion in air leading to the breakdown of $N_2$ and $O_2$, and $NO_x$ production via the Zeldovitch mechanism), where road traffic is the predominant source in many urban areas (Keuken *et al.,* 2009; Grice *et al.,* 2009; Carslaw and Rhys-Tyler, 2013). Consequently, boundary layer $NO_x$ abundance varies over many orders of magnitude – from sub 5-ppt levels in the remote marine boundary layer, to ppm levels in some urban environments (Crawford *et al.,* 1997).

Techniques used for the measurement of atmospheric $NO_x$ include laser-induced fluorescence spectroscopy (LIF), for both NO and $NO_2$; absorption spectroscopy (long-path and cavity-enhanced differential optical absorption spectroscopy, LP- and CE-DOAS, cavity attenuated phase shift spectroscopy (CAPS) and passive diffusion tubes, primarily for $NO_2$), chemical ionisation mass spectrometry (CIMS) and both on- and offline wet chemical methods *e.g.* long path absorption photometer (LOPAP) (Heard, 2008; Sandholm et al. 1990; Kasyutich et al. 2003; Kebabian et al. 2005; Cape, 2009; Fuchs et al. 2009; Villena et al. 2011). However, the most commonly employed technique for the measurement of $NO_x$ species, including for statutory air quality monitoring purposes, is the detection of the chemiluminescence arising from electronically excited $NO_2$ ($NO_2^*$) formed from the reaction between NO and $O_3$ (via R1):

$$NO + O_3 \quad \rightarrow \quad NO_2^* + O_2 \qquad (R1)$$
$$NO_2^* \quad \rightarrow \quad NO_2 + h\nu \qquad (R2)$$





Chemiluminescent instruments mix sampled ambient air with a reagent stream containing an excess of
ozone, to promote the chemiluminescent reaction; the resulting emission signal is measured using a
photomultiplier tube (PMT), and consists of contributions from $NO_2^*$ formed as above, but also
potentially from other chemiluminescence processes, detector dark counts and other noise
contributions. Contributions to the measured emission from other species are minimised by using a red
filter on the detector to block emission wavelengths below ca. 600 nm, and by employing a background
subtraction cycle: chemiluminescent $NO_x$ monitors commonly acquire a background by increasing the
reaction time between NO (from the sampled air) and $O_3$ (reagent formed within the instrument), using
a pre-reactor volume, such that nearly all of the NO present (specifications typically state, in excess of
99%) is converted to $NO_2$. The difference in PMT signals between the "online" and "background"
signals is then taken to be proportional to the NO present in the air sample, following the assumption
that the abundance of other species which may contribute to the measured signal is not affected by the
background cycle.

Chemiluminescent instruments typically alternate between two operation modes – one that directly
measures NO and one that measures $\Sigma(NO + NO_2)$, by first converting $NO_2$ to NO. The difference
between the two values determines the $NO_2$ mixing ratio (if only NO and $NO_2$ are present). This is most
commonly achieved using a molybdenum (Mo) catalyst heated to $300 - 350°C$. However, the reduction
of other $NO_z$ species to NO have led to the use of these catalysts in chemiluminescent $NO_y$ monitors to
measure total reactive nitrogen rather than $NO_2$ ($NO_y = NO_z + NO_x$; and $NO_z$ = other reactive nitrogen
species catalysed by Mo convertors *e.g.* $HNO_3$, HONO, $N_2O_5$, $HO_2NO_2$, PAN, $NO_3$, organic nitrates –
but not $NH_3$) (Navas *et al.,* 1997; Murphy *et al.,* 2007). If atmospheric mixing ratios of $NO_z$ species are
high relative to $NO_2$ then $NO_2$ measurements with monitors equipped with Mo catalysts are increasingly
inaccurate. This has led to the adoption of photolytic $NO_2$ conversion stages in research instruments,
where a blue light LED convertor is illuminated in a photolysis cell converting $NO_2$ to NO (Lee *et al.,*
118   2015).


120                $NO_2 + h\nu\ (\leq 395\ nm)\quad\rightarrow\quad NO + O(^3P)$          (R3)


The photolytic conversion technique can have greater specificity than the heated Mo catalyst as the
photolysis wavelengths may be selected to match the $NO_2$ photolysis action spectrum, while potential
$NO_z$ interferents for an $NO_2$ measurement are thermally unstable and may convert to $NO_2$ when exposed
to heat in the latter approach (Heard, 2008). Despite this, the chemiluminescent analyser with the heated
Mo catalyst is the most widely used technique for air quality monitoring of NO and $NO_2$ worldwide. It
is the reference method of measurement specified in the EU directive (BS EN 14211, 2012), providing
real-time data with short time resolution for 212 monitoring sites, including kerbside, roadside, urban
background, industrial and rural locations (Air Quality Expert Group, 2004).

*Origins of interferences in chemiluminescent $NO_x$ measurements*

While $NO_x$ measurements are sometimes perceived to be straightforward and routine, in practice a
number of factors are known to affect the accuracy of the levels obtained using chemiluminescence
approaches. A detailed account of factors affecting atmospheric $NO_x$ measurement overall is given
elsewhere (*e.g.* Gerboles *et al.*, 2003; Villena *et al.*, 2012; Reed *et al.,* 2016); here we do not focus upon
surface sources/losses but rather upon chemical interferences in chemiluminescent $NO_x$ analysers,
which may arise from the following possible general mechanisms:





1.  Collisional quenching of $NO_2^*$ by an interferent species with a greater collisional efficiency than the bath gas (e.g. air) used for calibration (typically a negative interference, although the magnitude and sign of this depends upon the calibration conditions employed)
2.  Conversion of other N-containing species to $NO_x$ within the $NO_2$ conversion unit (positive interference)
3.  Chemical removal or interconversion of NO and/or $NO_2$ by an interferent species generated within the instrument (positive or negative interference)
4.  Chemiluminescence of other chemical species, which is not fully accounted for during the instrument background cycle (positive interference)

Collisional quenching of excited species, mechanism (1), results in a reduction in the chemiluminescence intensity, to an extent dependent upon the pressure, and quenching efficiency – the efficacy with which the quenching species may accept or remove energy from the excited moiety. In the case of electronically excited $NO_2$, effective quenching agents have been shown to include $H_2O$, $CO_2$, $H_2$ and hydrocarbons (Matthews *et al*., 1977; Gerboles *et al*., 2003; Dillon and Crowley, 2018), of which only quenching by water vapour is considered to be significant under most common (ambient air) conditions – sensitivity reductions of up to 8 % have been reported (Steinbacher *et al*., 2007). Mechanism (2), conversion of other nitrogen-containing species to NO, alongside $NO_2$, is a recognised issue with heated Mo converters – interferences between 18 – 100 % have been reported for species such as HONO, $HNO_3$, PAN, alkyl nitrates and $N_2O_5$ (Dunlea *et al*., 2007; Lamsal *et al*., 2008). To address these uncertainties, photolytic converters are now commonly employed in research measurements, although for most routine air quality monitoring, heated Mo converters are still employed. Recently, it has been shown that a further interference can arise within the photolytic converter stage – from the generation of $HO_x$ radicals through photolysis of photolabile carbonyl species such as glyoxal, forming peroxy radicals promoting NO-to-$NO_2$ conversion within the instrument (Villena *et al*., 2012), resulting in a negative $NO_2$ interference, which may (under some conditions) exceed the positive interference from retrieval of $NO_z$ species associated with heated Mo converter instruments i.e. mechanism (3).

The focus of this work relates to mechanism (4): interference in chemiluminescent measurements of NO and $NO_2$ (using both catalytic and photolytical converters) arising from the chemiluminescence of alkenes in the presence of ozone. Alkene-ozone reactions have received substantial attention as a dark source of $HO_x$ radicals, and a route to the formation of semi-volatile compounds which contribute to secondary organic aerosol (SOA), particularly for biogenic alkenes such as isoprene and the mono- and sesquiterpenes (*e.g.* Johnson & Marston, 2008; Shrivastava *et al.,* 2017). Rate constants for ozonolysis reactions depend on alkene structure, and are typically larger for biogenic alkenes. Chemiluminescence from ozonolysis reactions was first reported by Finlayson *et al*. (1974), and indeed has been used to perform field measurements of both ozone and alkenes (e.g. Velasco *et al*., 2007; Hills and Zimmerman, 1990). This combination – of alkene-ozone reactions giving rise to a chemiluminescent interference signal, and alkene-ozone reactions being sufficiently rapid that alkenes can be appreciably consumed in the background (pre-reactor) cycle, and hence the interference contribution not fully subtracted during the background correction – gives rise to the potential for interference in $NO_x$ measurements, which is the focus of this study.



## EXPERIMENTAL APPROACH

Experiments were performed using chamber A of the two 200m$^3$ simulation chambers of the European Photoreactor (EUPHORE) facility in Valencia, Spain to provide a common, homogeneous air volume for multiple NO$_x$ analysers to sample from. The EUPHORE chambers are formed from fluorine-ethene-propene (FEP) Teflon foil fitted with housings that exclude ambient light (Wiesen, 2001; Munoz *et al.*, 2011). The chambers are fitted with large horizontal and vertical fans to ensure rapid mixing (timescale 3 minutes). Instrumentation used comprised long-path FTIR (for absolute and specific alkene / VOC measurements), monitors for temperature, pressure, humidity (dew-point hygrometer), ozone (UV absorption) and CO (infrared absorption). NO$_x$ levels were measured using four independent chemiluminescent monitors, plus (in the case of NO$_2$) LP-DOAS absorption spectroscopy – All monitor sampling lines were attached to one inlet sampling from the centre of the chamber.

Monitors 1 and 2 employed heated Mo catalysts, while 3 and 4 used photolytic NO$_2$ converters (see Table 1). All NO$_x$ monitors were calibrated at the start and end of the two-week measurement period using a multi-point calibration derived from a primary NO standard (BOC 5ppm alpha standard, certified to the NPL scale) in addition to single-point calibrations performed on a daily basis. NO$_2$ calibration was achieved via gas-phase titration using added ozone within the chamber. In some experiments the calibrations and interference were confirmed with use of the EUPHORE long-path DOAS system to unequivocally identify and quantify NO$_2$.

All experiments were performed with the chamber housing closed (i.e. dark conditions, $j(NO_2) < 2 \times 10^{-6} \, s^{-1}$), at near atmospheric pressure and ambient temperature. For most experiments, humidity was low (dew point ca. -45 ° C). The experimental procedure, starting with a clean flushed chamber, was to add SF$_6$ (as a dilution tracer), followed by successive aliquots of various alkenes and in certain cases additional species (H$_2$O and CO), whilst recording the measured NO and NO$_2$ levels, over periods of 1-3 hours. For some systems, ozone was added at the end of the experiment – under such dark, high O$_3$ conditions we can be confident that negligible NO could actually be present in the chamber (e.g. from wall sources) and hence that any "NO" signal observed by the monitors was unequivocally an interference response (as any NO remaining would be rapidly consumed by reaction with O$_3$). The potential interferant species investigated were cis-2-butene (C2B), trans-2-butene (T2B), tetra-methyl ethylene (2,3-dimethyl-butene or TME), α-terpinene, limonene, methyl chavicol (estragole) and terpinolene, with 4 – 5 additions of 20 – 50 ppb in each case, together with single- or dual-point interference measurements for ethene, propene, isobutene, isoprene, α-pinene, β-pinene and myrcene. Repeat experiments were performed for trans-2-butene, terpinolene and α-terpinene under conditions of increased humidity (up to ca. 30% RH). Alkene mixing ratios introduced into the chamber are given in Table S1. Propene, cis-2-butene and trans-2-butene where supplied by The Linde Group (purity > 99%); isobutene (purity > 99%) and terpinolene (purity > 85%) from Fluka Analytical; and TME (purity > 98%), isoprene (purity > 99%), limonene (purity > 97%), α-pinene (purity > 97%), β-pinene (purity > 97%), α-terpinene (purity > 85%), estragole (purity > 98%) and myrcene (purity > 99%) from Sigma Aldrich. All reagents were used as supplied.

### *Data Analysis*

The limit of detection (LOD) for each instrument was determined under the actual experimental conditions, as three times the standard deviation of the NO and NO$_2$ signal recorded each day from the empty chamber prior to the start of experiments (*i.e.* before addition of any reactants). The mean LODs





determined for NO and NO$_2$ are shown in Table 1. These LOD values are higher than those quoted by
the manufacturers for monitors 1-4 (typically 2-100 ppt) but accurately reflect the actual performance
of the instruments as used during these experiments. In the analysis which follows, in order to confirm
that any change in measured NO and NO$_2$ mixing ratio for each alkene addition was not due to noise or
drift and therefore signal, the readings were compared to the experimentally determined LOD for each
instrument. Only if the measured change was greater than the experimentally determined LOD were
these readings used for determining an interference. The interference due to the VOC was determined
by means of linear regression (least squares fit), with slopes and their uncertainty and Pearson's
correlation coefficients calculated in IGOR (see Tables 2 and 3).
**RESULTS**
Figures 1-3 give the measured VOC mixing ratios and the retrieved "NO" and "NO$_2$" measurements by
the four monitors during the experiment for selected alkenes, along with the regression analysis for
determining the interference levels. Spikes in NO and NO$_2$ mixing ratios observed after an alkene
addition (*e.g.* Figure 3) arise from sampling close to the addition point prior to the initial period of
mixing in the chamber (~ 3 min) and were disregarded in the analysis. The slow decay of alkene and
"NO$_x$" mixing ratios following each addition arises from dilution effects (~ $5.7 \times 10^{-5}$ s$^{-1}$, derived from
the decay of SF$_6$).
From Figures 1-3, a clear and systematic response from the monitors to the presence of α-terpinene,
terpinolene and trans-2-butene was observed, with the magnitude varying between the monitors. In
addition to the alkenes in Figures 1-3, significant interference effects were also observed for cis-2-
butene, TME and limonene for some of the monitors, as summarised in Tables 2 and 3. No interference
was observed, within detection uncertainty, for ethene, propene, isobutene, α-pinene, β-pinene,
myrcene or methyl chavicol in any of the monitors. For isoprene, no statistically significant interference
was observed for monitors 1-3, while monitor 4 observed a very small positive interference of 0.035 ±
0.001% (NO channel) and 0.076 ± 0.002% (NO$_2$ channel).
For the alkenes where significant interference was observed, in general a positive interference was
observed for NO and a negative interference for NO$_2$ by monitors 1-4 (Tables 2 and 3), with the
exception of TME, where a negative NO interference was observed by monitor 3 (and is discussed
later). Generally, for monitor 4 a positive NO interference, and a mixture of both positive and negative
NO$_2$ interferences, was observed. Overall, while the magnitude of interference differed between the
monitors, the same trend in the interference was observed, with α-terpinene having the largest
interference effect, followed by terpinolene, TME/trans-2-butene, cis-2-butene and limonene.
The addition of water (RH ca. 30%) led to the observed NO and NO$_2$ interference for trans-2-butene,
terpinolene and α-terpinene decreasing by 30 – 60% as shown in Tables 2 and 3. The addition of CO
resulted in an increase in the NO interference observed for TME from below the LOD to 0.7% for
monitors 1 and 2 while monitors 3 and 4 exhibited a larger interference increase (Table 2).
**DISCUSSION**
*Interference effects on retrieved NO abundance*



Positive NO interferences were observed for those alkenes which react most rapidly with ozone, and
hence will be present within the monitor reaction chamber at different levels in the measurement and
background modes.  This interference is attributed to chemiluminescent emission following the alkene-
ozone reaction, and may be attributed to a combination of two factors: formation of excited products in
the alkene-ozone reaction which emit chemiluminescence, coupled with the significant removal of some
alkenes during the instrument background phase compared with the measurement phase, through their
reaction with (elevated levels of) ozone within the instrument, *i.e.* mechanism (4) outlined above.
Possible origins of this signal are the production of excited HCHO, vibrationally excited OH and
electronically excited OH (*e.g.* Finlayson *et al*., 1974).   While the long-pass filters used in
chemiluminescence $NO_x$ monitors should preclude emission from electronically excited species,
vibrationally excited OH produced through the hydroperoxide mechanism is known to emit in the 700
– 1100 nm wavelength range (Finlayson *et al.,* 1974; Schurath *et al.,* 1976; Hansen *et al.,* 1977; Toby,
1984), and would be detected as $NO_2$.
The difference in the interference effect among monitors may then reflect differences in the conditions
(ozone abundance, pressure, residence time) within the reaction cell and filter specifications. The
relative magnitudes of the positive interference signals observed between the different monitors are
consistent with this picture, as the reaction chamber pressure is much lower for monitors 3 and 4 (*ca.* 1
– 10 Torr) compared with monitors 1 and 2 (*ca.* 300 Torr) leading to greater collisional quenching.
Similarly, addition of $H_2O$, which would be expected to efficiently accept vibrational energy from OH
radicals (Gerboles *et al*., 2003), was found to substantially reduce the apparent interference. In the
experiments with higher humidity, a reduced interference (factor of *ca.* 2, see Table 2) was observed
for all NO experiments for all instruments except for TME for the photolytic converters, where an
increase was observed. There is currently no recommended relative humidity in which calibrations
should be performed for any of the instruments or within EU and EPA guidelines (AQEG, 2004;
USEPA, 2002). However, the installation of permeation driers at the sample inlet should (in principle)
reduce the impact of different $H_2O$ / relative humidity levels upon quenching of $NO_2$ or other species
and are a common feature of most modern samplers (AQEG, 2004).
*Interference magnitude: kinetic and structural effects*
The most significant effects are the large positive NO interferences observed for the monoterpenes; α-
terpinene and terpinolene, within monitors 1, 3 and 4. The criteria for an alkene to display such a
positive interference (*i.e.* via mechanism 4) is that it reacts with ozone to produce suitable excited
products which exhibit a chemiluminescent signal at appropriate wavelengths. In addition, the alkene
must have a sufficiently rapid reaction with ozone that its mixing ratio is substantially reduced during
the instrument background phase compared with the measurement phase, precluding the correct
subtraction of the interference signal.  The reaction rate constants for many alkenes with ozone are well
known, allowing the calculation of a kinetic interference potential (KIP) ranking for this second factor
(see Supplementary Information for calculation details). The calculated KIP are shown in Table 4 as
the percentage of a given alkene's potential chemiluminescent signal which would *not* be subtracted in
the standard background cycle, under the assumption that the background cycle conditions ($O_3$ mixing
ratio, residence time) would be sufficient to remove 99% of NO present.
This ranking does not reflect the precise (relative) interference which is observed, as it neglects
structural features which will affect the product yield (and state *i.e.* electronic or vibrationally excited)





of the chemiluminescent products from the ozonolysis reaction – but is consistent with the trend and
relative magnitudes for the substantial positive interferences shown in Tables 2 and 3. For example, a
lack of interference is observed for myrcene and limonene, both of which exhibit terminal C=C bonds
(see Table 4), and after reaction with ozone lead to the production of the $CH_2OO$ Criegee intermediate
(CI) which subsequently decomposes or undergoes rearrangement to form small yields of OH (Alam *et*
*al.*, 2011). The ozonolysis of internal alkenes such as cis- and trans-2-butene produce the $CH_3CHOO$
CI which predominantly decomposes via the vinyl hydroperoxide mechanism forming larger yields of
OH (Johnson and Marston, 2008; Alam *et al.*, 2013). Such chemically formed OH that produces a
detectable signal may also be augmented by contributions from $HO_2$ and $RO_2$, converted into OH within
the instrument by reaction with NO – especially in the $NO_2$ channel of photolytic converter instruments.

The relationship between the KIP (Table 4) and measured NO interference (Tables 2 and 3) is illustrated
in Figure 4 and can be used for predicting the potential interference of a given alkene to the NO signal
form a kinetic perspective. For example, α-humulene has a KIP of 94.54% which could give rise to a
1.7%, 2.4% or 10.2% NO interference for monitors 1, 3 and 4, respectively. This is, however, based on
the rate constant of α-humulene alone and does not include any structural features such as the presence
of terminal and non-terminal C=C bonds.


*Explanation of the interference observed for $NO_2$*

The above discussion considers only the interference effect arising from alkene chemiluminescent
emission; further measurement impacts are also evident in the (negative) interferences apparent for
other species / monitors in Tables 2 and 3. Inspection of Tables 2 and 3 shows smaller positive
interferences, and some negative interferences, from alkenes in the $NO_2$ measurements.

$NO_2$ measurements using chemiluminescence approaches are usually obtained by measuring $NO_x$ (*i.e.*
$\Sigma(NO + NO_2)$, after passing the sampled air through an $NO_2$ converter) and subtracting the
(independently determined) NO contribution. If the actual interference signal (additional
chemiluminescence) during the $NO_x$ measurement mode arises solely from mechanism (4), ozonolysis
chemiluminescence, then this would be expected to match that in the NO mode (subject to the alkene
abundance not being altered in the $NO_2$ conversion stage and if the detection conditions for the NO and
$NO_x$ phases are identical), and consequently would not affect the retrieved $NO_2$ mixing ratio. Monitors
1, 2 and 3 used a single detection cell, alternating between NO and $NO_2$ ($NO_x$) modes, and so measured
the $NO_2$* chemiluminescence signal under identical conditions (optical arrangement, filtering,
pressure). The observed negative interference for $NO_2$ therefore may have arisen due to removal of
alkene by the Mo catalyst within the monitors.


For monitor 1 (TE 42i-TL), the negative interference observed for $NO_2$ was the same magnitude as
observed for the positive interference for NO, including the experiments with $H_2O$ and CO (see Figure
5 and Tables 2-3). This response is thought to arise as a consequence of the calculation methodology,
combined with removal of alkenes during the $NO_2$ conversion by the Mo catalyst:

There are three modes of operation in monitor 1 (TE 42i-TL) – NO measurement, $NO_2/NO_x$
measurement and background (pre-reactor) measurement, given by Eq 1-3 respectively:





$$sNO = sNO_{real} + X_i \qquad \text{(Eq 1)}$$

$$sNOx = sNOx_{real} + yX_i \qquad \text{(Eq 2)}$$

$$sP = fX_i \qquad \text{(Eq 3)}$$


where $sNO$ and $sNOx$ are the NO and $NO_x$ signals produced by the chemiluminescence monitor,
respectively, $sNO_{real}$ and $sNOx_{real}$ are the 'real' NO and $NO_x$ signals, $X_i$ denotes the interference
alkene $i$, $y$ is the fraction of the interferant (alkene) $X_i$ remaining after the Mo convertor, $sP$ denotes
signal at the pre-reactor and $f$ is the fraction of $X_i$ remaining after the pre-reactor. The mixing ratios of
NO, $NO_2$ and $NO_x$ are given by:

$$[NO] = \frac{sNO - sP}{cNO} \qquad \text{(Eq 4)}$$

$$[NO] = \frac{(sNO_{real} + X_i) - fX_i}{cNO} \qquad \text{(Eq 5)}$$

$$[NO] = \frac{(sNO_{real} + (1-f)X_i)}{cNO} \qquad \text{(Eq 6)}$$


$$[NOx] = \frac{sNOx - sP}{cNOx} \qquad \text{(Eq 7)}$$

$$[NOx] = \frac{(sNOx_{real} + yX_i) - fX_i}{cNOx} \qquad \text{(Eq 8)}$$

$$[NOx] = \frac{(sNOx_{real} + (y-f)X_i)}{cNOx} \qquad \text{(Eq 9)}$$


$$[NO_2] = \frac{[NO_x] - [NO]}{CE} \qquad \text{(Eq 10)}$$

$$[NO_2] = \frac{(sNOx_{real} + (y-f)X_i)}{cNOx \times CE} - \frac{(sNO_{real} + (1-f)X_i)}{cNO \times CE} \qquad \text{(Eq 11)}$$


where $c$ is the 'span factor' and $CE$ represents the conversion efficiency. If we assume $cNOx \approx cNO \approx$
$c$, then

$$[NO_2] = \frac{(sNOx_{real} + (y-f)X_i) - (sNO_{real} + (1-f)X_i)}{c \times CE} \qquad \text{(Eq 12)}$$




From Eq 12, it may be seen that if $y = 1$ (*i.e.* if the interferant – alkene – abundance is not affected by
passage through the Mo converter), then there would be no interference observed in the retrieved $NO_2$,
while if the interferant species is subject to removal during passage through the converter, then $y < 1$
and a negative interference would be observed. Molybdenum oxide catalysts have been reported to
efficiently isomerise alkenes at temperatures between $300 – 400\ °C$, (Wehrer *et al.,* 2003) and are also
effective catalysts for the epoxidation of alkenes (Shen *et al.,* 2019). The observed small negative
interference effects (for monitors 1 and 2, the Mo converter units), in the absence of significant sampled
$NO_x$, may reflect partial removal of the alkene on the converter.
The negative $NO_2$ interference apparent for monitors 3 and 4 (photolytic converter instruments) is more
difficult to rationalise (as no Mo catalyst is present). Under ambient conditions, where $NO_x$ is present,
mechanism (3) may occur as outlined below. In reality, the conversion efficiency for photolytic
converters is substantially lower than 100% (Reed et al. 2016), as a consequence of both the finite
photolysis intensity achievable, and occurrence of the $NO + O_3$ back reaction. If the instrument
calibration factor for $NO_x$ is not equal to that for NO (see Eq 11), or if alkene was removed in the
convertor stage, then this will lead to different interferences for NO and $NO_2$, as CE is also
(significantly) less than 1. This trend is apparent in the values shown in Table 3, in particular for the
instruments fitted with photolytic convertors. However, in the absence of sampled $NO_x$ the observed
less-positive or even negative $NO_2$ interference suggests that less alkene is present in the $NO_x$ mode.
Direct photolysis of alkenes is unlikely to cause such a change, considering the photolytic converter
wavelength envelope, but photolytic production of $HO_x$ radicals (which then react with the alkene) may
be responsible.
Monitor 4 (AQD) used independent $NO_2^*$ detection channels; tests were conducted using both channels
for cis-2-butene and terpinolene systems, and revealed significant differences between the two detectors
(*ca.* 40% lower interference response for NO in the $NO_2$ detection channel). With two independent
detection channels, $NO_2$ may be determined from the $NO_x$ measurement by either subtracting the NO
level obtained from the NO channel (method (a)), or via the difference in signal observed in the
$NO_2/NO_x$ channel when turning the photolysis lamp on and off (method (b)). Under method (a), as
employed for cis-2-butene and terpinolene, a lower positive interference from alkene
chemiluminescence results, as a consequence of the difference in the detection cell conditions (results
marked * in Table 3), while under method (b), as employed for the other alkenes studied here with the
AQD system, the interference (from mechanism 4 alone) should cancel out (results marked # in Table
424 3).

*Effect of quenching by the alkenes*
The data presented in Figures 1-3 and Tables 2 and 3 show both negative and positive interferences
while mechanism 4 alone would be expected to result in positive interference signals for NO for all
alkenes. We therefore conclude that additional mechanisms are occurring. Under the conditions of
these chamber experiments, retrieval of additional $NO_y$ species can be precluded (the chamber wall
source of HONO has been characterised and shown to produce ppt levels of HONO under the dark, dry
conditions of these experiments (Zador *et al*., 2005) and would be equally present for all experiments).
We attribute the negative (or reduced positive) interference effects to a combination of mechanisms (1)
and (3): quenching of excited OH (produced by alkene+ozone reaction) by alkenes – electron rich
alkenes have been shown to be effective quenchers (Gersdorf *et al.,* 1987; Chang and Schuster, 1987)
- and generation of $HO_x$ radicals within the instrument following on from the ozonolysis reaction.



The alkene-ozone reactions are known to produce OH, $HO_2$ and $RO_2$ radicals both directly (e.g. Johnson
and Marston, 2008), following the photolysis of other alkene-ozone reaction products (e.g. carbonyl
compounds), and through OH-alkene reactions. Peroxy radicals promote the conversion of NO to $NO_2$,
altering the abundance of both species (the formation of $NO_x$ reservoirs such as nitric acid and organic
nitrates will also occur, but will be negligible on the timescale of operation of most instruments).
The ozonolysis of TME results in the production of OH with close to unity yield (IUPAC, 2018) and if
taking into account the above mechanism (4) only, might be expected to exhibit a large interference in
NO mode. Table 2 shows no interference for monitors 1 and 2 (Mo convertor units) and negative and
positive interferences for monitors 3 and 4 (photolytic convertor units) respectively, and so is hard to
rationalise (for NO mode). The addition of CO as a scavenger for OH led to an increase in the NO signal
for all monitors. A possible origin of this signal is the production of the excited intermediate HOCO
(from reaction of vibrationally excited OH, from the ozonolysis of TME, with CO), which has a
temperature and pressure dependent rate of reaction, (Atkinson *et al.,* 2006; Li and Francisco, 2000)
and is consistent with the larger NO signal in the photolytic monitors (Table 2).
**CONCLUSION**
The interference in chemiluminescence $NO_x$ measurements from alkenes has been systematically
investigated using four commercially available monitors. There are varying degrees of interferences in
the NO and $NO_2$ signals by all monitors investigated and are due to a combination of mechanisms 1, 3
and 4. Monoterpenes, α-terpinene and terpinolene, exhibit the largest interferences followed by 2,3-
dimethyl-2-butene (TME) and trans-2-butene, in line with the calculated KIP (see Table 4). The KIP
can be used as a crude indicator for a potential interference of an alkene to a NO signal, but have large
margins of error as they do not take into account the variation in the yield of chemiluminescent products
and other instrumental differences. The alkene interference observed with enhanced RH conditions also
indicates the need to accurately calibrate chemiluminescence $NO_x$ analysers under actual sampling
conditions.
The NO interferences from alkenes among the monitors investigated in this study ranges from 1 to 11%.
The varying responses exhibited by the different monitors reflect differences in the conditions within
the instrument (ozone abundance, pressure and residence time) within the reaction cell and filter
specifications. The magnitude of the NO and $NO_2$ interferences not only vary with different alkenes
and commercial monitors, but will also be dependent upon sampling environments (and with trends in
ambient $NO_x$ and alkenes). Notably, in these experiments the alkene abundance is high compared with
most ambient air samples – consequently internally generated OH will react essentially exclusively with
the alkene, which may not reflect ambient sampling – but which we do not expect to impact the
conclusions reached with respect to mechanism 4, interference in retrieved NO levels. Further research
to explore these impacts, and other parameters (*e.g.* $H_2O$ abundance), is urgently needed.
Mixing ratios of $NO_x$ vary from > 100 ppb in some urban areas, *e.g.* Marylebone Road (Carslaw *et al.*
2005), < 300 ppt in biogenic environments (Hewitt *et al.* 2010) and < 35 ppt in remote areas (Lee et al.
2009). For typical urban environments where alkene mixing ratios are relatively low (< 2 ppb e.g. von
Schneidemesser *et al.* 2010) these interferences are not likely to be significant (~ 1% of the NO signal).
However, for biogenic environments where monoterpenes and sesquiterpenes, which react rapidly with
ozone, are abundant, this interference could be significantly larger. For example, average mixing ratios
for isoprene (~ 1 ppb), 5 monoterpenes (~ 220 ppt), 3 short chain alkenes (~ 240 ppt) and NO (0.14
ppb) were measured within a south-east Asian tropical rainforest (Jones *et al.,* 2011). Using the





relationship between KIP and NO interference an overestimation of NO levels of to up to 58% may be
observed, with very significant implications for prediction of other atmospheric chemical processes
involving $NO_x$. Given that $NO_x$ mixing ratios are relatively small in biogenic and remote environments,
these interferences could lead to a substantial overestimation. Alkene interference contribute to the
relatively high NO and low $NO_2$ reported in the tropical rainforest at night, which could not be otherwise
accounted for (Pugh *et al*. 2011).
Within indoor environments, $NO_x$ primarily arises from outdoor sources or indoor combustion sources
(Young *et al.,* 2019). Typically, in the absence of a known indoor combustion source, indoor NO levels
are low (*ca.* 13% of outdoor levels) with $NO_2$ comprising the majority of the $NO_x$ (Zhou *et al.,* 2019). ,
There are multiple sources of alkene indoors, such as fragranced volatile personal care products
(Nemafollahi *et al.,* 2019; Yeoman *et al.,* 2020) and cleaning products (Kristenson *et al.,* 2019),
resulting in very much larger levels than $NO_x$ (McDonald *et al.,* 2018; Kristenson *et al.,* 2019).
Consequently, monoterpenes are among the most ubiquitous VOC reported for indoor air, with the main
species including, linalool, α-pinene, β-myrcene and limonene (Krol et al 2014; Nematollahi et al 2019).
Monoterpene mixing ratios in indoor environments are reported to be 5 to 7 times larger than those
reported outdoors (low ppb levels), and can be further enhanced by cleaning activities (Singer *et al.,*
2006; Kristenson *et al.,* 2019; Weschler and Carslaw, 2018). Peak limonene mixing ratios may be a
factor of *ca.* 50 higher indoors than outdoor environments (Colman Lerner *et al.,* 2012), while indoor
α-terpinene and α-pinene mixing ratios have exceeded 10 and 68 ppb, respectively (Singer *et al.,* 2006;
Brown *et al.,* 1994). These relatively large monoterpene ratios may lead to substantial interferences in
chemiluminescence $NO_x$ monitors; their incorrect retrieval as measured "$NO_x$" will impact assessments
of indoor air quality and health.
**DATA AVAILABILITY.**
Experimental data will be available in the Eurochamp database, www.eurochamp.org, from the
H2020 EUROCHAMP2020 project, GA no. 730997
**AUTHOR CONTRIBUTIONS**
MSA, WJB and JDL conceived and planned the experiments. MSA, JDL, MV, AM and MR performed
the experiments. LRC, LJK and MSA performed the data analysis. LRC, LJK, MSA, CF and WJB
contributed to data investigation and curation. MSA wrote the original draft manuscript and all co-
authors contributed to reviewing and editing the paper.
**COMPETING INTERESTS**
The authors declare that they have no conflict of interest.
**ACKNOWLEDGEMENTS**
This work was funded in part through the UK Natural Environment Research Council (NERC) project
"ICOZA: Integrated Chemistry of Ozone in the Atmosphere" (NE/ K012169/1) and by the
EUROCHAMP-2 Transnational access project "NOxINT: NOx analyser interference in chemically
complex mixtures" (E2-2010-05-26-0033) . Part of this work has received funding from the European
Union's Horizon 2020 research and innovation programme through the EUROCHAMP-2020
Infrastructure Activity under grant agreement No. 730997. CEAM is partly supported by the
IMAGINA-Prometeo project (PROMETEO2019/110) and by Generalitat Valenciana. In addition, we
thank Eva Clemente for their work in these experiments.





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





***Table 1:*** *Details of the NO$_x$ monitoring instruments used.*

| Number | Manufacturer | Model | Institution | NO$_2$ Convertor | Limit of Detection (LOD)* | |
|---|---|---|---|---|---|---|
| | | | | | NO (ppt) | NO$_2$ (ppt) |
| **1** | Thermo | TE42i-TL | Birmingham | Heated Mo | 210 | 210 |
| **2** | API | 200AU | EUPHORE | Heated Mo | 190 | 450 |
| **3** | Eco Physics | CLD 770 Alppt / PLC 760 | EUPHORE | Xe lamp | 150 | 430 |
| **4** | Air Quality Designs | - | York | Blue light at 395 nm | 60 | 150 |

*Calculated in this study






**Table 2:** *Measured NO interference (% ±1 s.d. of the slope) for each monitor across a range of different alkenes (LOD: Limit of Detection).*

| Species | 1: TE 42i-TL | 2: API 200AU | 3: Eco Physics CLD770 | 4: Air Quality Designs |
|---|---|---|---|---|
| cis-2-butene | < LOD | < LOD | 0.4 ± 0.05 | 0.38 ± 0.004 |
| TME | < LOD | < LOD | -0.7 ± 0.09 | 1.1 ± 0.001 |
| Trans-2-butene | < LOD | < LOD | 1.0 ± 0.008 | 0.83 ± 0.01 |
| Terpinolene | 0.5 ± 0.05 | < LOD | 1.3 ± 0.01 | 4.4 ± 0.15 |
| $\alpha$-Terpinene | 1.9 ± 0.05 | 0.5 ± 0.04 | 2.3 ± 0.04 | 10.9 ± 0.06 |
| Limonene | < LOD | < LOD | < LOD | -0.10 ± 0.001 |
| TME + $H_2O$ | < LOD | < LOD | 0.6 | 2.4 |
| Trans-2-butene + $H_2O$ | < LOD | < LOD | 0.48 ± 0.006 | 0.37±0.01 |
| Terpinolene + $H_2O$ | 0.25 ± 0.03 | < LOD | 0.88 ± 0.004 | 1.6 ± 0.1 |
| $\alpha$-Terpinene + $H_2O$ | 1.0 ± 0.07 | < LOD | 1.3 ± 0.06 | 6.2 ± 0.7 |
| TME + CO | 0.70 ± 0.002 | 0.66 ± 0.09 | 1.3 ± 0.12 | 1.4 ± 0.02 |




































**Table 3:** *Measured NO$_2$ interference (% $\pm$ 1 s.d. of the slope) for each monitor across a range of*
*different alkenes (LOD: Limit of Detection).*

| Species | 1: TE 42i-TL | 2: API 200AU | 3: Eco Physics CLD770 | 4: Air Quality Designs |
|---|---|---|---|---|
| cis-2-butene | -0.6 ± 0.1 | < LOD | -1.1 ± 0.08 | 0.3 ± 0.02 |
| TME | -0.63 ± 0.05 | < LOD | -0.78 ± 0.15 | -0.92 ± 0.1 |
| Trans-2-butene | -0.5 ± 0.06 | < LOD | -0.5 ± 0.03 | -0.93 ± 0.02 |
| Terpinolene | -0.61 ± 0.02 | < LOD | -0.18 ± 0.03 | 1.6 ± 0.1 |
| α-Terpinene | -1.9 ± 0.13 | < LOD | -1.0. ± 0.2 | 3.1 ± 2.1 |
| Limonene | < LOD | < LOD | < LOD | 0.09 ± 0.003 |
| TME + H$_2$O | -0.6 | < LOD | < LOD | -2.0 |
| Trans-2-butene + H$_2$O | < LOD | < LOD | < LOD | -0.41 ±0.02 |
| Terpinolene + H$_2$O | -0.29 ± 0.02 | < LOD | < LOD | -0.25 |
| α-Terpinene + H$_2$O | -0.98 ± 0.06 | < LOD | < LOD | 0.35±0.1 |
| TME + CO | -0.70±0.01 | < LOD | < LOD | 1.0 ± 0.3 |
































***Table 4:*** *Kinetic ranking of interference potential: the percentage of the potential chemiluminescent*
*signal from ozonolysis of a given alkene which would* not *be removed by a standard instrument*
*background cycle, under conditions (ozone mixing ratio, residence time) which would remove 99% of*
*the NO sampled.  Rate constants are taken from Calvert et al. (2000).  NB: this ranking does not include*
*variations in the yield of chemiluminescent products with alkene structure, which will modulate the*
*values given.  Species marked * are investigated in this study.*

| Species | $k_{(Alkene+O3)}$ (298 K) /cm$^3$ molecule$^{-1}$ s$^{-1}$ | Kinetic Interference Potential (%) | No. of C=C bonds | No. of terminal C=C bonds |
|---|---|---|---|---|
| Ethene | $1.58 \times 10^{-18}$ | 0.04 * | 1 | 1 |
| 1-Butene | $9.64 \times 10^{-18}$ | 0.23 | 1 | 1 |
| 2,3-dimethyl-1-butene | $1.00 \times 10^{-17}$ | 0.24 | 1 | 1 |
| Propene | $1.01 \times 10^{-17}$ | 0.24 * | 1 | 1 |
| 1-pentene | $1.06 \times 10^{-17}$ | 0.26 | 1 | 1 |
| Isobutene | $1.13 \times 10^{-17}$ | 0.27 * | 1 | 1 |
| Isoprene | $1.28 \times 10^{-17}$ | 0.31 * | 1 | 1 |
| 2-methyl-1-butene | $1.30 \times 10^{-17}$ | 0.31 | 1 | 1 |
| β-pinene | $1.50 \times 10^{-17}$ | 0.36 * | 1 | 1 |
| α-cedrene | $2.80 \times 10^{-17}$ | 0.68 | 1 | 0 |
| 3-carene | $3.70 \times 10^{-17}$ | 0.89 | 1 | 0 |
| α-pinene | $8.66 \times 10^{-17}$ | 2.08 * | 1 | 0 |
| cis-2-butene | $1.25 \times 10^{-16}$ | 2.98 * | 1 | 0 |
| cis-3-hexane | $1.44 \times 10^{-16}$ | 3.43 | 1 | 0 |
| trans-3-hexane | $1.57 \times 10^{-16}$ | 3.73 | 1 | 0 |
| α-coapene | $1.58 \times 10^{-16}$ | 3.76 | 1 | 0 |
| trans-2-butene | $1.90 \times 10^{-16}$ | 4.50 * | 1 | 0 |
| Limonene | $2.00 \times 10^{-16}$ | 4.73 * | 2 | 1 |
| 2-carene | $2.30 \times 10^{-16}$ | 5.42 | 1 | 0 |
| 2-methyl-2-butene | $4.03 \times 10^{-16}$ | 9.31 | 1 | 0 |
| Myrcene | $4.70 \times 10^{-16}$ | 10.77 * | 3 | 2 |
| 2,3-dimethyl-2-butene | $1.13 \times 10^{-15}$ | 23.96 * | 1 | 0 |
| Terpinolene | $1.90 \times 10^{-15}$ | 36.90 * | 2 | 0 |
| α-humulene | $1.20 \times 10^{-14}$ | 94.54 | 3 | 0 |
| β-carophyllene | $1.20 \times 10^{-14}$ | 94.54 | 2 | 1 |
| α-terpinene | $2.10 \times 10^{-14}$ | 99.38 * | 2 | 0 |







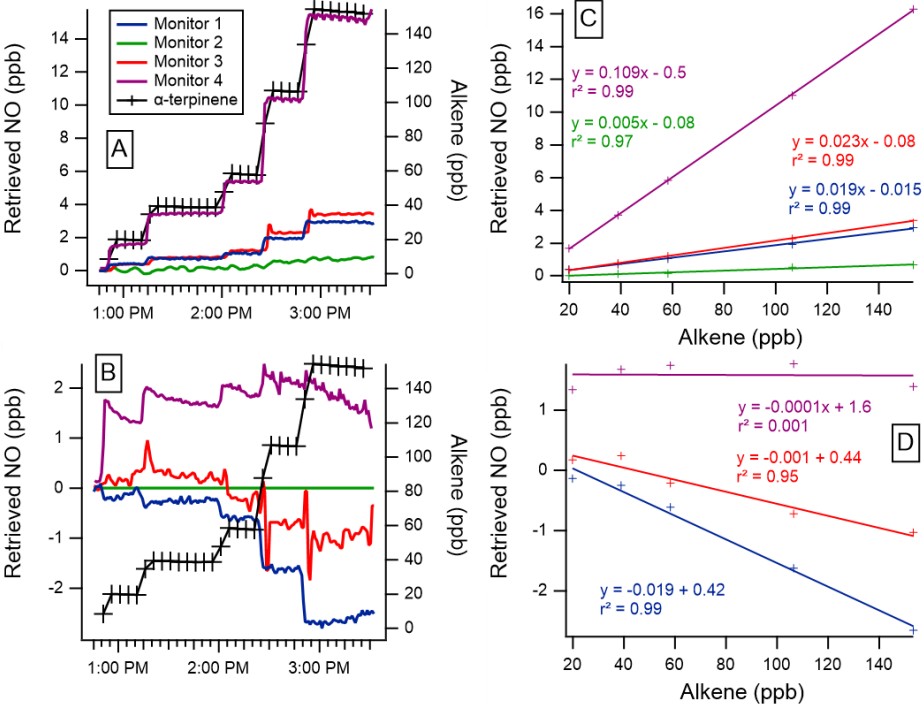

**Figure 1:** Time series of the α-terpinene mixing ratio and indicated / "measured" NO (top) and $NO_2$
(bottom) mixing ratios as directly retrieved by each monitor (left column) and the regression
calculations for the monitors that demonstrated significant interference with the addition of α-
terpinene (right column).  Note the different y-axis scales.



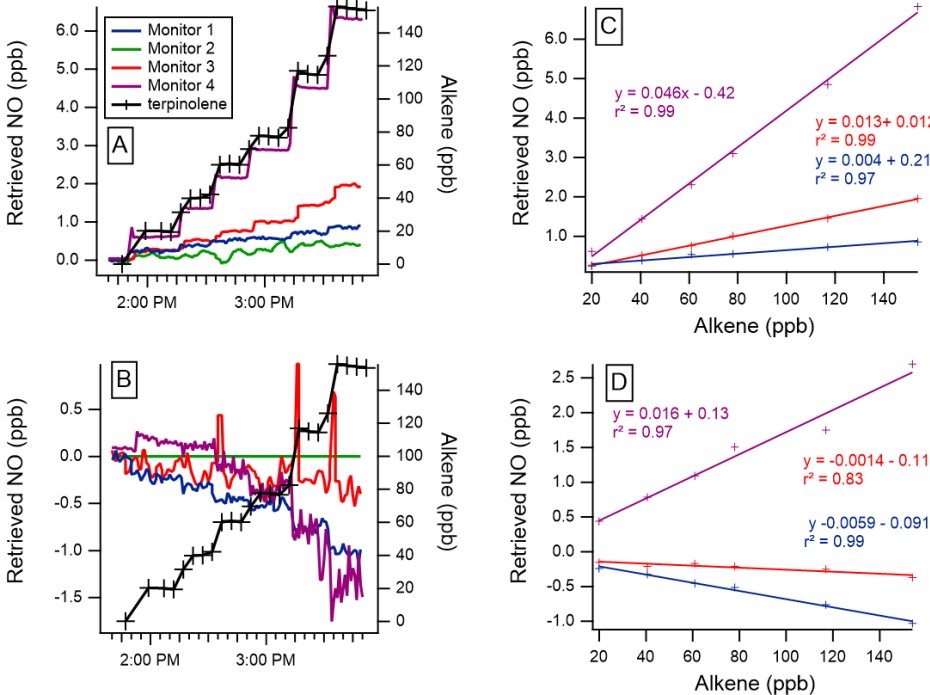

**Figure 2:** Time series of the terpinolene mixing ratio and measured NO and $NO_2$ mixing ratios as
retrieved by each monitor (left column) and the regression calculations for the monitors that
demonstrated significant interference with the addition of terpinolene (right column). Note the
different y-axis scales.


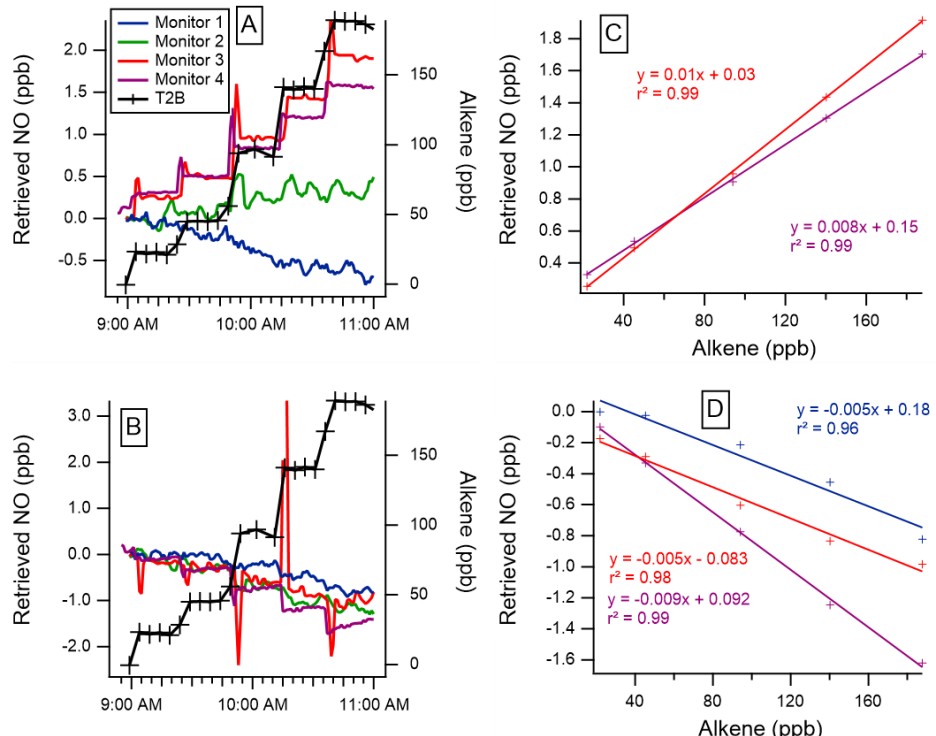

**Figure 3:** Time series of the trans-2-butene (T2B) mixing ratio and measured NO (top) and NO$_2$
(bottom) mixing ratios as retrieved by each monitor (left column) and the regression calculations for
the monitors that demonstrated significant interference with the addition of T2B (right column).
Note the different y-axis scales.




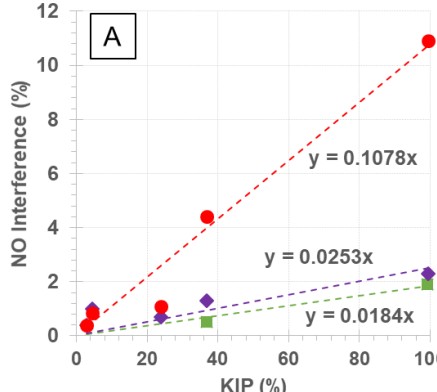 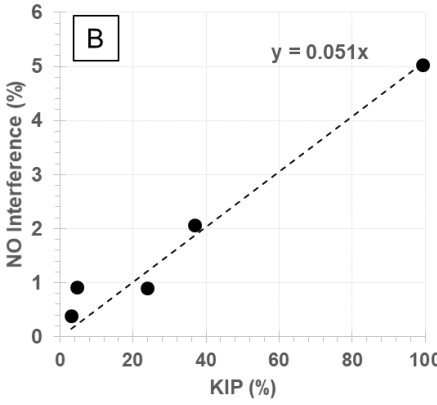

**Figure 4:** Relationship between measured NO interference (%) and kinetic interference potential, KIP
(%) for monitors 1 (green), 3 (purple), 4 (red) and the average of the observed NO interference
across all instruments (black).




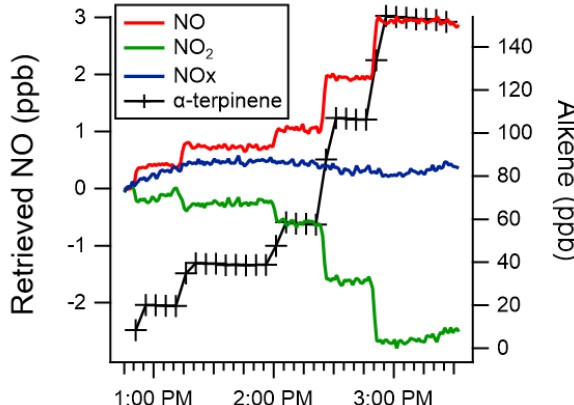

**Figure 5:** Time series of the $\alpha$-terpinene mixing ratio (black) and measured NO (red), NO$_2$ (green) and
NO$_x$ (blue) mixing ratios as retrieved by monitor 1 (TE 42i-TL).