# Peer review of "Interference from alkene in chemiluminescent NOx measurements"

_Atmospheric Measurement Techniques, 2020_

## Referee Comment (RC1) · Anonymous Referee #2 · 9 Jul 2020

The manuscript by Alam et al. presents a chamber study about the interference of alkenes in chemiluminescent NOx measurements. Varies of alkenes are studied and shown that the interference to NO ranged from 1% to 11%. However, the interference to NO2 detection is more complicated. Overall, this paper presented a useful study for promoting the high precision NOx measurement. Some comments should be addressed before considering the publication in AMT.

General comments. 1. The introduction of these NOx instruments should be added to the experimental section. I suggest the authors add a schematic figure to introduce the background and sampling mode of the NOx measurement, which could help the non-professional readers follow the background interference part easily. 2. Line 320, the KPI is a good indicator and easy to understand, but the Supplementary Information

for calculation details seems not finished as there is no equation of KPI = ???. Considering that the KPI is important in this paper, the final equation should be listed in the main text. 3. The NO measurement by monitor 2 has small interference by alkene, and NO2 measured by monitor 2 free of the interference of alkenes, does this result mean the API 200 AU monitor has better instrumental design compared with other monitors, at least in avoiding the alkene interference? 4. According to the results in table 2 and Line 258-259, monoterpenes have no interference. While in the conclusion part (Line 485 and 502), the author proposed the monoterpene should be noted, it is contradicted, please clarify it. 5. What happened about the monitor 2 in figure 1-2 in NO2 measurement? 6. Figure 1-3 is very confused. Why are some fitting results not shown? If the non-significant result not shown, why the measured NO2 by Monitor 4 is plotted in figure 1 with very r2=0.001?

Specific comments. 7. Line 79-85, the cited reference Fuchs et al., (2009) is about cavity ring-down spectroscopy, so the citation is wrong (also cavity-enhanced absorption spectroscopy should be mentioned). An appropriate reference should be added about CE-DOAS. 8. Line 203, missed a blank between 5 and ppm. There also many errors like this (e.g., Line 190...) 9. Line 296, the O3 abundance, and residence time are not discussed in the following paragraph. 10. Figure 1-3, panel B and D, change the y-axis as NO2 rather than NO (although the mixing ratio are retrieved as NO). 11. The average results in figure 4(B) do not make sense. I suggest removing it. 12. Figure 5, the left and right y-axis should be changed, please change to (NO/NO/NOx) and (ïĄą-terpinene). 13. The time resolution of data for the four monitors and shown in figures should be clarified. 14. Line 421-424, the label * and # are missed in Table 3. 15. The caption of Table 4 should add the reaction rate constant of NO+O3 (298 K) for intercomparison. 16. Line 450-453, are you mean the possible HOCO is an interference of the chemiluminescent?

---

## Referee Comment (RC2) · Anonymous Referee #1 · 14 Jul 2020

Interference from alkenes in chemiluminescent NOx measurements M.S. Alam et al., Atmospheric Measurement Techniques Discussions, doi:10.5194/amt-2020-164

The authors present a study of NO and NO2 measurements made in the presence of a series of alkenes in the EUPHORE atmospheric simulation chamber. Measurements of NO and NO2 were made using four instruments based on detection of chemiluminscence of excited NO2* formed by the reaction of NO with O3 generated within the instrument. This technique enables the direct measurement of NO, but measurements of NO2 require conversion of NO2 to NO, followed by measurement of the resulting NO which represents the sum of NO and NO2 concentrations and gives the concentration of NO2 from the difference between the sum of NO and NO2 and the measurement of NO alone. Two of the instruments used in this study use catalytic conversion of NO2

to NO using a heated Mo catalyst, while the other two instruments employ photolytic conversion using a blue LED.

The authors outline a number of potential interferences in NOx measurements that can affect instruments based on detection of chemiluminescence, and primarily focus on potential chemical interferences resulting from detection of chemiluminescence from species other than NO2*. Given the importance of accurate NOx measurements for air quality, the results of this study are potentially significant. The experimental procedures seem robust, the paper is well-written and within the scope of the journal, and will be of interest to the wider atmospheric chemistry community. However, there are a number of areas which should be improved in the manuscript prior to publication.

In general, the discussion of the observed effects is somewhat limited and the manuscript would benefit from expanding the possible causes of the interference and providing some recommendations for future experiments to identify and eliminate interferences as far as possible. Several species are mentioned as being potentially responsible for the chemiluminscence interference, including excited HCHO, vibrationally excited OH and electronically excited OH. Some discussion of the filters used in the NOx instruments is given, but it would be informative, where possible, to give the emission spectra of possible interfering species, NO2*, and the filters used in the instruments employed in this study. Are there significant differences between filters in different instruments? Could future work using alternative filters rule out interferences from these species? Could emission spectra of the chemiluminescence interferences be measured in future experiments?

Some discussion of the kinetics of ozone-alkene reactions is given in comparison to the observed interferences, which indicates that more rapid ozone-alkene reactions are more likely to result in interferences. Consideration of the energetics of the ozone-alkene reactions investigated, combined with modelling of the chemistry involved, might be more insightful and could help to identify whether production of excited species is likely, and which species with appropriate emission spectra might be
present in sufficient concentration to produce significant interferences.

Minor comments are given below.

Lines 54-55: This sentence appears to be incomplete.

Line 128: Are the 212 monitoring sites in the UK, EU or a wider area?

Line 154: Are the CO2 mixing ratios in the chamber elevated significantly above ambient levels such that interferences could result in the chamber?

Lines 175-178: Please provide further details of the previous work. What alkenes were investigated? What were the conditions? Were emission spectra reported? If so, what were the emission wavelengths? Do the previous studies give any further details on which species might have been responsible for the observed chemiluminescence?

Line 199: Were the sampling lines all of similar length?

Line 203: What were the concentration ranges over which calibrations were performed?

Line 280: Can the relationship between the level of interference and the alkene + ozone reaction rate be quantified in any way? Does a plot of the level of interference against rate of reaction reveal any general trend?

Lines 295-296: What are the differences in conditions between instruments?

Line 332: Is CH2OO the only possible Criegee intermediate produced? What other species/Criegee intermediates are produced?

Line 364: Is there any likely effect of the age of the catalyst?

Line 496: Remove the comma at the end of the line.

Table 2: Values and uncertainties should be given to the same number of significant figures.

[Figure]

Figures 1, 2, & 3: Panels B & D should be labelled as NO2 on the y-axes.

[Figure]

---

## Author Comment (AC1) · 15 Sep 2020

We would like to thank the reviewer for their comments and suggestions, and for taking the time to review our manuscript.

"The manuscript by Alam et al. presents a chamber study about the interference of alkenes in chemiluminescent NOx measurements. Varies of alkenes are studied and shown that the interference to NO ranged from 1% to 11%. However, the interference to NO2 detection is more complicated. Overall, this paper presented a useful study for promoting the high precision NOx measurement. Some comments should be addressed before considering the publication in AMT.

General comments.

[Figure]

1. The introduction of these NOx instruments should be added to the experimental section. I suggest the authors add a schematic figure to introduce the background and sampling mode of the NOx measurement, which could help the non-professional readers follow the background interference part easily."

RESPONSE: We think that the introduction to chemiluminescent NOx instruments (line 71 – 81) is more suited for the introduction and have not moved it to the experimental section. This is because: (i) this information is generic and not specific to our experimental set up, and (ii) in order to understand the potential origins of interferences in chemiluminescent NOx monitors, the knowledge of a typical instrument setup is required. A schematic diagram (Fig 1 – see below) has been added to this section to help the non-professional readers, as suggested by the reviewer. This has been referred to in Line 73.

"2. Line 320, the KPI is a good indicator and easy to understand, but the Supplementary Information for calculation details seems not finished as there is no equation of KPI = ???. Considering that the KPI is important in this paper, the final equation should be listed in the main text." RESPONSE: The final KIP expression has been included into the main text in Lines 259 – 267. The detailed calculation of the KIP remains unchanged in the Supplementary Information.

"3. The NO measurement by monitor 2 has small interference by alkene, and NO2 measured by monitor 2 free of the interference of alkenes, does this result mean the API 200 AU monitor has better instrumental design compared with other monitors, at least in avoiding the alkene interference?" RESPONSE: The data presented in this study indicates that the API 200 AU monitor instrument responds least to alkene interference.

"4. According to the results in table 2 and Line 258-259, monoterpenes have no interference. While in the conclusion part (Line 485 and 502), the author proposed the monoterpene should be noted, it is contradicted, please clarify it." RESPONSE: $\alpha$-

[Figure]

Terpinene (C10H16), terpinolene (C10H16) and limonene (C10H16) are all monoterpenes. The results shown in Table 2 show the largest interferences from $\alpha$-terpinene and terpinolene both of which are monoterpenes. In lines 258-259 (now 206 – 209) we do not report all monoterpenes in having no NO interference, but report the response of individual alkenes / monoterpenes that do not exhibit an interference within the detection limits of the instruments. The conclusion has been amended to remove any contradictory messages by including the following sentence: "Although monoterpenes, $\alpha$-pinene, myrcene and limonene, show no significant NO interferences in chemiluminscence NOx monitors, other fast reacting monoterpenes (with O3) such as $\alpha$-terpinene and terpinolene which are not generally reported in the literature, exhibit large interferences and may lead to substantial overestimations in NOx measurements." This is found in Lines 422 – 429.

"5. What happened about the monitor 2 in figure 1-2 in NO2 measurement?" RESPONSE: The NO2 measurements for monitor 2 in Figure 1-2 were zero throughout the experiment measurement period. There were no indications that there was anything wrong with the instrument before, during and after the experiment(s). The manuscript remains unchanged.

"6. Figure 1-3 is very confused. Why are some fitting results not shown? If the non-significant result not shown, why the measured NO2 by Monitor 4 is plotted in figure 1 with very r2=0.001?" RESPONSE: We are only meant to show the fittings that were significant. The reviewer is correct to point out that the NO2 measured by monitor 4 is not significant. This has been amended in the Figures.

"Specific comments.

7. Line 79-85, the cited reference Fuchs et al., (2009) is about cavity ring-down spectroscopy, so the citation is wrong (also cavity-enhanced absorption spectroscopy should be mentioned). An appropriate reference should be added about CE-DOAS." RESPONSE: Added "cavity ring-down spectroscopy (CRDS)" – Line 61. Reference

also added: "Thalman, R., and Volkamer, R.: Inherent calibration of a blue LED-CE-DOAS instrument to measure iodine oxide, glyoxal, methyl glyoxal, nitrogen dioxide, water vapour and aerosol extinction in open cavity mode. Atmos. Meas. Tech., 3, 1797-1814, 2010." See lines 625 – 627 in the reference list.

"8. Line 203, missed a blank between 5 and ppm. There also many errors like this (e.g., Line 190. . .)" RESPONSE: Amended. Line 61 and anywhere where we use units.

"9. Line 296, the O3 abundance, and residence time are not discussed in the following paragraph." RESPONSE: The intention of this paragraph was to discuss the differences in interference magnitudes due to the varying pressures within the reaction chamber of the different instruments. This has been clarified by the addition of "e.g." in line 240. Ozone (reagent formed within the instrument) specifications typically state in excess abundance, in order to convert all (or >99%) NO present into NO2. Increasing the reaction time between the NO (from sampled air) and excess O3 would allow more time for NO to be converted into NO2. This is explained in the introduction in lines 71 – 81.

"10. Figure 1-3, panel B and D, change the y-axis as NO2 rather than NO (although the mixing ratio are retrieved as NO)." RESPONSE: Amended – See Figures 2-4 in manuscript.

"11. The average results in figure 4(B) do not make sense. I suggest removing it." RESPONSE: This is an average of the interferences calculated across all instruments vs KIP%. This allows us to calculate the relative potential interference response from any monitor from a given alkene rather than an absolute upper limit (for monitor 4) or lower limit (for monitor 1) from this study only. We think including this figure allows the community to calculate relative potential interferences from other monitors. The manuscript remains unchanged.

"12. Figure 5, the left and right y-axis should be changed, please change to

(NO/NO/NOx) and (ï ËŻA ËŻa-terpinene)." RESPONSE: Amended (see Fig. 6 in manuscript).

"13. The time resolution of data for the four monitors and shown in figures should be clarified." RESPONSE: All figures shown use 1 minute time resolution data for all monitors. This is included in the caption for Figures 2, 3, 4 and 6 in the manuscript.

"14. Line 421-424, the label * and # are missed in Table 3." RESPONSE: The data in Table 3 have been labelled with * and #.

"15. The caption of Table 4 should add the reaction rate constant of NO+O3 (298 K) for intercomparison." RESPONSE: "k(NO+O3)= 1.90 × 10-14 cm3 molecule 1 s-1 (298 K)" has been added to the caption of Table 4.

"16. Line 450-453, are you mean the possible HOCO is an interference of the chemilu-minescent?" RESPONSE: Yes. We have amended the manuscript to explain this more clearly by adding the word "chemiluminescence". See line 373-376.
* * *
[Figure]

[Figure]

**Fig. 1.** A typical flow schematic of a chemiluminescent NO monitor.

---

## Author Response (AR1)

**Interference from alkene in chemiluminescent NOx measurements**

[revised manuscript text omitted]

$$NO + O_3 \quad\quad \rightarrow \quad\quad NO_2^* + O_2 \quad\quad\quad\quad\quad\quad\quad\quad\quad\quad\quad\quad\quad\quad\quad (R1)$$

$$NO_2^* \quad\quad\quad \rightarrow \quad\quad NO_2 + h\nu \quad\quad\quad\quad\quad\quad\quad\quad\quad\quad\quad\quad\quad\quad\quad (R2)$$

The intensity of the light emitted via (R2) is in the wavelength 600 – 3000 nm, peaking at ~1200 nm. Chemiluminescent instruments mix sampled ambient air with a reagent stream containing an excess of ozone, to promote the chemiluminescent reaction (see schematic – Figure 1); the resulting emission signal is measured using a photomultiplier tube (PMT), and consists of contributions from $NO_2^*$ formed as above, but also potentially from other chemiluminescence processes, detector dark counts and other noise contributions. Contributions to the measured emission from other species are minimised by using a red filter on the detector to block emission wavelengths below ca. 600 nm, and by employing a background subtraction cycle: chemiluminescent $NO_x$ monitors commonly acquire a background by increasing the reaction time between NO (from the sampled air) and $O_3$ (reagent formed within the instrument), using a pre-reactor volume, such that nearly all of the NO present (specifications typically state, in excess of 99%) is converted to $NO_2$. The difference in PMT signals between the "online"

and "background" signals is then taken to be proportional to the NO present in the air sample, following the assumption that the abundance of other species which may contribute to the measured signal is not affected by the background cycle.

Chemiluminescent instruments typically alternate between two operation modes – one that directly measures NO and one that measures $\Sigma(NO + NO_2)$, by first converting $NO_2$ to NO. The difference between the two values determines the $NO_2$ mixing ratio (if only NO and $NO_2$ are present). This is most commonly achieved using a molybdenum (Mo) catalyst heated to 300 – 350°C. However, the reduction of other $NO_z$ species to NO have led to the use of these catalysts in chemiluminescent $NO_y$ monitors to measure total reactive nitrogen rather than $NO_2$ ($NO_y = NO_z + NO_x$; and $NO_z$ = other reactive nitrogen species catalysed by Mo convertors *e.g.* $HNO_3$, HONO, $N_2O_5$, $HO_2NO_2$, PAN, $NO_3$, organic nitrates – but not $NH_3$) (Navas *et al.,* 1997; Murphy *et al.,* 2007). If atmospheric mixing ratios of $NO_z$ species are high relative to $NO_2$ then $NO_2$ measurements with monitors equipped with Mo catalysts are increasingly inaccurate. This has led to the adoption of photolytic $NO_2$ conversion stages in research instruments, where a blue light LED convertor is illuminated in a photolysis cell converting $NO_2$ to NO (Lee *et al.,* 2015).

$$NO_2 + h\nu \ (\leq 395 \text{ nm}) \quad\quad \rightarrow \quad\quad NO + O(^3P) \quad\quad\quad\quad\quad\quad\quad\quad\quad\quad\quad\quad (
[revised manuscript text omitted]

**Interactive comment on "Interference from alkenes in chemiluminescent NOx measurements"**

**by Mohammed S. Alam et al.**

Anonymous Referee #1

Interference from alkenes in chemiluminescent NOx measurements M.S. Alam et al.,

Atmospheric Measurement Techniques Discussions, doi:10.5194/amt-2020-164

The authors present a study of NO and NO2 measurements made in the presence of a series of alkenes in the EUPHORE atmospheric simulation chamber. Measurements of NO and NO2 were made using four instruments based on detection of chemiluminscence of excited NO2* formed by the reaction of NO with O3 generated within the instrument. This technique enables the direct measurement of NO, but measurements of NO2 require conversion of NO2 to NO, followed by measurement of the resulting NO which represents the sum of NO and NO2 concentrations and gives the concentration of NO2 from the difference between the sum of NO and NO2 and the measurement of NO alone. Two of the instruments used in this study use catalytic conversion of NO2 to NO using a heated Mo catalyst, while the other two instruments employ photolytic conversion using a blue LED.

The authors outline a number of potential interferences in NOx measurements that can affect instruments based on detection of chemiluminescence, and primarily focus on potential chemical interferences resulting from detection of chemiluminescence from species other than NO2*. Given the importance of accurate NOx measurements for air quality, the results of this study are potentially significant. The experimental procedures seem robust, the paper is well-written and within the scope of the journal, and will be of interest to the wider atmospheric chemistry community. However, there are a number of areas which should be improved in the manuscript prior to publication.

In general, the discussion of the observed effects is somewhat limited and the manuscript would benefit from expanding the possible causes of the interference and providing some recommendations for future experiments to identify and eliminate interferences as far as possible. Several species are mentioned as being potentially responsible for the chemiluminscence interference, including excited HCHO, vibrationally excited OH and electronically excited OH. Some discussion of the filters used in the NOx instruments is given, but it would be informative, where possible, to give the emission spectra of possible interfering species, NO2*, and the filters used in the instruments employed in this study. Are there significant differences between filters in different instruments? Could future work using alternative filters rule out interferences from these species? Could emission spectra of the chemiluminescence interferences be measured in future experiments?

**RESPONSE:** *Some recommendations for future experiments to identify and eliminate interferences have been added to the conclusion in lines 396 – 400. "Further research to explore these impacts, and other parameters (e.g. $H_2O$ abundance), is urgently needed. The chemiluminescence from monoterpene ozonolysis should also be investigated to identify emission spectra of possible interfering species; given the varying OH yields and energetics from the ozonolysis of different alkenes, the intensity of emission are likely to vary. A combination of selective long-pass filters and detector characteristics can then be exploited within chemiluminescence $NO_x$ monitors to eliminate such interferences with similar emission spectra to $NO_2^*$."*

*The long-pass filters used in the chemiluminescence $NO_x$ monitors in this study are not reported in their respective user manuals, but typically block light below ca. 600 nm, while typical PMT response characteristics are between 400 – 950 nm. Any chemiluminescence signal in the 600 – 950 nm wavelength range can therefore cause a potential interference. This has been added to the text in lines 235 – 238. The emission wavelengths of potential interferents: excited HCHO, vibrationally excited OH and electronically excited OH have been given also in the introduction (lines 140 – 143) as requested by referee #2.*

Some discussion of the kinetics of ozone-alkene reactions is given in comparison to the observed interferences, which indicates that more rapid ozone-alkene reactions are more likely to result in interferences. Consideration of the energetics of the ozone-alkene reactions investigated, combined with modelling of the chemistry involved, might be more insightful and could help to identify whether production of excited species is likely, and which species with appropriate emission spectra might be present in sufficient concentration to produce significant interferences.

**RESPONSE:** *We agree that this would be interesting to study, however, modelling the chemistry involved to predict whether the responsible excited species would be present in concentrations likely to cause significant interferences would be out of the scope of this study. Modelling these types of experiments to predict excited species and their emission spectra would be a potential further study. Further research that is needed has been added to the manuscript in lines 396 – 400.*

Minor comments are given below.

Lines 54-55: This sentence appears to be incomplete.
**RESPONSE:** *Amended. Line 37 – added "because"*

Line 128: Are the 212 monitoring sites in the UK, EU or a wider area?
**RESPONSE:** "In the UK" added to Line 101.

Line 154: Are the CO2 mixing ratios in the chamber elevated significantly above ambient levels such that interferences could result in the chamber?

**RESPONSE:** The $CO_2$ mixing ratios within the chamber are not elevated significantly above ambient, and are therefore unlikely to affect the interference results. The manuscript remains unchanged.

Lines 175-178: Please provide further details of the previous work. What alkenes were investigated? What were the conditions? Were emission spectra reported? If so, what were the emission wavelengths? Do the previous studies give any further details on which species might have been responsible for the observed chemiluminescence?

**RESPONSE:** *This information has been added into the text. See line 139 – 142. "Chemiluminescence from the ozonolysis of*
*14 short chain alkene species at total pressures of 2 – 10 Torr was first reported by Pitts et al. (1972). Excited HCHO, vibrationally excited OH and electronically excited OH in the wavelengths 350 – 520 nm, 700 – 1100 nm and 306 nm, respectively, were the identified chemiluminescent species (Finlayson et al., 1974); and indeed has been used to perform field measurements of both ozone and alkenes (e.g. Velasco et al., 2007; Hills and Zimmerman, 1990)."*

*Added reference to line 594-595: Pitts Jr, J.N., Kummer, W.A., Steer, R.P., and Finlayson, B.J.: The chemiluminescent*
*reactions of ozone with olefins and organic sulphides, Advances in Chemistry, 113, 10, 246-254, 1972.*

Line 199: Were the sampling lines all of similar length?
**RESPONSE:** Yes. This is added into the manuscript. Line 156

Line 203: What were the concentration ranges over which calibrations were performed?
**RESPONSE:** *The calibration range has been included into the text: "(in the range 0 – 100 ppb)". Line 160*

Line 280: Can the relationship between the level of interference and the alkene + ozone reaction rate be quantified in any way? Does a plot of the level of interference against rate of reaction reveal any general trend?
**RESPONSE:** *This is explained in detail in the discussion section. The relationship between the level of interference and $k_{(alkene+o3)}$ has been described in the "interference magnitude: kinetic and structural effects" section and the use of the kinetic interference potential (KIP). The manuscript remains unchanged.*

Lines 295-296: What are the differences in conditions between instruments?
**RESPONSE:** *The main differences between the instruments are the different pressures within the reaction chambers and the different NO to NO2 conversion technologies. This is explained in line 241 – 244 and 159 – 161, respectively. The manuscript remains unchanged.*

Line 332: Is CH2OO the only possible Criegee intermediate produced? What other species/Criegee intermediates are
produced?

**RESPONSE:** *Myrcene contains three C=C bonds, two of which are terminal bonds, while limonene possesses two C=C bonds, one of which is terminal. This information is given in Table 4. Ozonolysis of terminal C=C bonds will lead to a CH2OO CI (the simplest of CIs), while the ozonolysis of internal C=C bonds will results in different CI structures (dependent upon the alkene). Each CI resulting from an internal C=C bond (of a different alkene) will not only be different in structure but will*
*also have different yields depending on the energetics of the reaction. This discussion is not in the scope of the study and is not the primary focus of the paper – the manuscript remains unchanged.*

*The possible CI formed from limonene and myrcene ozonolysis can be found in the following studies:*

- *Deng, P., Wang, L. and Wang, L., 2018. Mechanism of gas-phase ozonolysis of β-myrcene in the atmosphere. The*
*Journal of Physical Chemistry A, 122(11), pp.3013-3020.*
- *Baptista, L., Pfeifer, R., da Silva, E.C. and Arbilla, G., 2011. Kinetics and thermodynamics of limonene ozonolysis. The Journal of Physical Chemistry A, 115(40), pp.10911-10919.*

Line 364: Is there any likely effect of the age of the catalyst?
**RESPONSE:** *To the authors knowledge there are no known effects of the age of the Mo catalyst in the $NO_x$ monitors, so this is unlikely to cause any deviation in the results presented. The largest interference observed were from the photolytic convertor NOx monitors (AQD and Eco Physics) which are not catalysts. The manuscript remains unchanged.*

Line 496: Remove the comma at the end of the line.
**RESPONSE:** *Amended. Line 417*

Table 2: Values and uncertainties should be given to the same number of significant figures.
**RESPONSE:** *Amended. All values have been amended to 3sf. See Tables 2 and 3.*

Figures 1, 2, & 3: Panels B & D should be labelled as NO2 on the y-axes.
**RESPONSE:** *Amended. See Figures 2, 3 and 4.*

**Interactive comment on "Interference from alkenes in chemiluminescent NOx measurements"**

**by Mohammed S. Alam et al.**

Anonymous Referee #2

The manuscript by Alam et al. presents a chamber study about the interference of alkenes in chemiluminescent NOx measurements. Varies of alkenes are studied and shown that the interference to NO ranged from 1% to 11%. However, the interference to NO2 detection is more complicated. Overall, this paper presented a useful study for promoting the high precision NOx measurement. Some comments should be addressed before considering the publication in AMT.

General comments.

1. The introduction of these NOx instruments should be added to the experimental section. I suggest the authors add a schematic figure to introduce the background and sampling mode of the NOx measurement, which could help the non-professional readers follow the background interference part easily.

**RESPONSE:** *We think that the introduction to chemiluminescent NOx instruments (line 71 – 81) is more suited for the introduction and have not moved it to the experimental section. This is because: (i) this information is generic and not specific to our experimental set up, and (ii) in order to understand the potential origins of interferences in chemiluminescent NOx monitors, the knowledge of a typical instrument setup is required. A schematic diagram (Figure 1) has been added to this section to help the non-professional readers, as suggested by the reviewer. This has been referred to in Line 73.*

2. Line 320, the KPI is a good indicator and easy to understand, but the Supplementary Information for calculation details seems not finished as there is no equation of KPI = ???. Considering that the KPI is important in this paper, the final equation should be listed in the main text.

**RESPONSE:** *The final KIP expression has been included into the main text in Lines 259 – 267. The detailed calculation of the KIP remains unchanged in the Supplementary Information.*

3. The NO measurement by monitor 2 has small interference by alkene, and NO2 measured by monitor 2 free of the interference of alkenes, does this result mean the API 200 AU monitor has better instrumental design compared with other monitors, at least in avoiding the alkene interference?

**RESPONSE:** *The data presented in this study indicates that the API 200 AU monitor instrument responds least to alkene interference.*

4. According to the results in table 2 and Line 258-259, monoterpenes have no interference. While in the conclusion part (Line 485 and 502), the author proposed the monoterpene should be noted, it is contradicted, please clarify it.

**RESPONSE:** *α-Terpinene ($C_{10}H_{16}$), terpinolene ($C_{10}H_{16}$) and limonene ($C_{10}H_{16}$) are all monoterpenes. The results shown in Table 2 show the largest interferences from α-terpinene and terpinolene both of which are monoterpenes. In lines 258-259 (now 206 – 209) we do not report all monoterpenes in having no NO interference, but report the response of individual alkenes / monoterpenes that do not exhibit an interference within the detection limits of the instruments. The conclusion has been amended to remove any contradictory messages by including the following sentence: "Although monoterpenes, α-pinene, myrcene and limonene, show no significant NO interferences in chemiluminscence $NO_x$ monitors, other fast reacting monoterpenes (with $O_3$) such as α-terpinene and terpinolene which are not generally reported in the literature, exhibit large interferences and may lead to substantial overestimations in $NO_x$ measurements." This is found in Lines 422 – 429.*

5. What happened about the monitor 2 in figure 1-2 in NO2 measurement?

**RESPONSE:** *The NO2 measurements for monitor 2 in Figure 1-2 were zero throughout the experiment measurement period. There were no indications that there was anything wrong with the instrument before, during and after the experiment(s). The manuscript remains unchanged.*

6. Figure 1-3 is very confused. Why are some fitting results not shown? If the non-significant result not shown, why the measured NO2 by Monitor 4 is plotted in figure 1 with very r2=0.001?

**RESPONSE:** *We are only meant to show the fittings that were significant. The reviewer is correct to point out that the NO2 measured by monitor 4 is not significant. This has been amended in the Figures 2.*

Specific comments.

7. Line 79-85, the cited reference Fuchs et al., (2009) is about cavity ring-down spectroscopy, so the citation is wrong (also cavity-enhanced absorption spectroscopy should be mentioned). An appropriate reference should be added about CE-DOAS.

**RESPONSE:** *Added "cavity ring-down spectroscopy (CRDS)" – Line 61. Reference also added: "Thalman, R., and Volkamer, R.: Inherent calibration of a blue LED-CE-DOAS instrument to measure iodine oxide, glyoxal, methyl glyoxal, nitrogen dioxide, water vapour and aerosol extinction in open cavity mode. Atmos. Meas. Tech., 3, 1797-1814, 2010." See lines 625 – 627 in the reference list.*

8. Line 203, missed a blank between 5 and ppm. There also many errors like this (e.g., Line 190. . .)

**RESPONSE:** *Amended. Line 61 and anywhere where we use units.*

9. Line 296, the O3 abundance, and residence time are not discussed in the following paragraph.

**RESPONSE:** *The intention of this paragraph was to discuss the differences in interference magnitudes due to the varying pressures within the reaction chamber of the different instruments. This has been clarified by the addition of "e.g." in line 240. Ozone (reagent formed within the instrument) specifications typically state in excess abundance, in order to convert all (or >99%) NO present into $NO_2$. Increasing the reaction time between the NO (from sampled air) and excess $O_3$ would allow more time for NO to be converted into $NO_2$. This is explained in the introduction in lines 71 – 81.*

10. Figure 1-3, panel B and D, change the y-axis as NO2 rather than NO (although the mixing ratio are retrieved as NO).

**RESPONSE**: *Amended – See Figures 2-4 in manuscript.*

11. The average results in figure 4(B) do not make sense. I suggest removing it.

**RESPONSE:** *This is an average of the interferences calculated across all instruments vs KIP%. This allows us to calculate the relative potential interference response from any monitor from a given alkene rather than an absolute upper limit (for monitor 4) or lower limit (for monitor 1) from this study only. We think including this figure allows the community to calculate relative potential interferences from other monitors. The manuscript remains unchanged.*

12. Figure 5, the left and right y-axis should be changed, please change to (NO/NO/NOx) and (ï ¸A ¸a-terpinene).

**RESPONSE:** *Amended (see Figure 6).*

13. The time resolution of data for the four monitors and shown in figures should be clarified.

**RESPONSE:** *All figures shown use 1 minute time resolution data for all monitors. This is included in the caption for Figures 2, 3, 4 and 6 in the manuscript.*

14. Line 421-424, the label * and # are missed in Table 3.

**RESPONSE:** *The data in Table 3 have been labelled with * and # as discussed in line 350-352.*

15. The caption of Table 4 should add the reaction rate constant of NO+O3 (298 K) for intercomparison.

**RESPONSE:** *"$k_{(NO+O3)}= 1.90 \times 10^{-14} cm^3 molecule^{-1} s^{-1}$ (298 K)" has been added to the caption of Table 4.*

16. Line 450-453, are you mean the possible HOCO is an interference of the chemiluminescent?

**RESPONSE:** *Yes. We have amended the manuscript to explain this more clearly. See line 373-376.*

**Additional changes to the manuscript**

**General**

 *We have formatted the manuscript according to the guidelines set for AMT/AMTD and as a result the line numbers may be slightly out of sync. We have noted this in the responses below.*

**Abstract**

*Line 11 – added "and air pollution"*

 *Line 13 – added "formed" and "within the NOx analyser"*

*Line 20/21 – deleted "the" and added "conventional", "s" to cycle and "that in"*

*Line 25 – deleted "Alkenes" and replaced with "The species"*

**Introduction**

 *Line 44 – replaced the word "abundance" with "concentration"*

**Results**

*Line 200 – added "with a first order rate constant of"*

*Line 204 – added "shown"*

 *Line 247 – replaced "in" with "under"*

**Discussion**

*Line 255 – replaced "is" with "are"*

*Line 271 – deleted "product"*

 *Line 284 – added "estimate"*

**Conclusion**

*Lines 379 – 384 – additional information has been added here*

*Line 404, 406, 409, 411 – 413, 417 – the sentences have been improved*

**Data Availability**

*Line 434 – replaced "will be" with "are"*